# Hierarchical-Concatenate Fusion TDNN for sound event classification

Baishan Zhao☯, Jiwen Liang☯* 

School of Information Science and Engineering, Shenyang University of Technology, Shenyang City, Liaoning Province, China

☯ These authors contributed equally to this work.
* a17741226289@163.com

**Data Availability Statement:** Access to the UrbanSound8K data set is available upon request. These pages contain information on how to get the data as well as direct links to the request forms: https://doi.org/10.5281/zenodo.1203745.

## Abstract

Semantic feature combination/parsing issue is one of the key problems in sound event classification for acoustic scene analysis, environmental sound monitoring, and urban soundscape analysis. The input audio signal in the acoustic scene classification is composed of multiple acoustic events, which usually leads to low recognition rate in complex environments. To address this issue, this paper proposes the Hierarchical-Concatenate Fusion (HCF)-TDNN model by adding HCF Module to ECAPA-TDNN model for sound event classification. In the HCF module, firstly, the audio signal is converted into two-dimensional time-frequency features for segmentation. Then, the segmented features are convolved one by one for improving the small receptive field in perceiving details. Finally, after the convolution is completed, the two adjacent parts are combined before proceeding with the next convolution for enlarging the receptive field in capturing large targets. Therefore, the improved model further enhances the scalability by emphasizing channel attention and efficient propagation and aggregation of feature information. The proposed model is trained and validated on the Urbansound8K dataset. The experimental results show that the proposed model can achieve the best classification accuracy of 95.83%, which is an approximate improvement of 5% (relatively) over the ECAPA-TDNN model.

## Section 1: Introduction

Sound Event Classification [1] (SEC) is to classify sound events in the real-world environment into predefined categories using a trained system. The complexity and diversity of acoustic events pose challenges for sound classification models. The research objective is to optimize the ECAPA-TDNN network model architecture to achieve higher accuracy and reliability in different acoustic events. In recent years, the recognition and classification of sound events have garnered increasing attentions from researchers in artificial intelligence. For instance, Rashid et al. [2] used the identification of cough sounds to aid in the preliminary screening for COVID-19. Moreover, SEC is of significance to a wide range of applications in domains such as music recognition, speech recognition, and environmental sound classification. Sound classification is the extraction of key features from audio signals. Commonly methods include Mel

**Funding:** The author(s) received no specific funding for this work.

**Competing interests:** The authors have declared that no competing interests exist.

spectrograms [3–5], MFCC [6, 7], log-Mel spectrograms [8–10], and Fbank [11, 12]. These algorithms facilitate the extraction of pertinent acoustic features from raw audio data.

In terms of SEC, researchers have introduced various deep learning models. For example, the ResNet, proposed by Kaiming in 2015 [13], solved degradation in Deep Neural Networks ((DNN)) [14–16]. Besides, the ResNet [17, 18] and its variants [19–21] are combined with the SE module [22–24] to enhance the robustness of sound classification model. With the rise of Convolutional Neural Networks (CNN)-based methods in various sound signal, such as TS-CNN10 [25], ResNeXt-GAP [26], and AST-P [27]. CNN Network Architecture have become the standard method for addressing various audio classification problems. For instance, Su et al. [28] proposed a TSCNN-DS model composed of two four-layer CNNs to extract features from the LMC feature set and MC feature set, followed by classification. Some methods are based on shallow CNN architectures [29, 30] and have obtained some results in SEC tasks. However, shallow CNN architectures are limited by their network depth in extracting local high dimensional audio features, and cannot focus on features in the time-frequency domain over longer distances. To improve CNN model performance in SEC, variants of CNN emerged prominently in 2021 [31]. VGG-M [32, 33], with a complexity between VGG16 [34, 35] and VGG19 [36, 37], demonstrates good results in speaker recognition. Yet, its original design for image processing made it less adept at capturing essential features from sequential sound data. Furthermore, prevalent neural network architectures are based on Time Delay Neural Networks (TDNN) [38], which is widely used in Automatic Speaker Verification, such as x-vector system. For instance, the renowned ECAPA-TDNN network architecture is proposed by Desplanques et al. from the Ghent University in 2020 [39], which achieves a low Equal Error Rate (EER) of 0.95%. However, this model overlooks capturing large receptive fields to identify significant targets. Motivated by the limitations of capturing long-range dependencies, several professionals have recently adopted the use of attention mechanisms to address the sound classification problem. For example, AF-TDNN [40] built upon the ECAPA-TDNN and provided a novel attention mechanism. While there are improvements in performance, the focus remained on refining the extraction capabilities for smaller targets. In a similar study, Sharma et al. [41] proposed an ADCNN-5 model composed of CNN and attention blocks based on DCNN and used a four-channel feature atlas composed of MFCC, GFCC. Subsequent models like SMMT model [42], have combined attention mechanisms with spiking neural network to emphasize certain partial features to further improve performance. Although it enlarges the perceptual field, increases the depth and ability of feature extraction of the network, the number of parameters is much larger than that of DNN. The network model is also too deep to cause degradation and other problems more easily. In audio processing-related tasks, feature correlation over long distances cannot be ignored. To obtain more discriminative and powerful information, Li et al. [43] proposed CAR-Transformer Neural Network Model. Besides, Transformer-based models [44, 45] like Conformer [46] and Efficient Conformer [47] algorithms based on Transformer network are verified the effectiveness in text recognition in the last three years. However, their generalization performances when handling various acoustic properties remain a consideration. As shown in Table 1, the three key performance metrics for sound event classification problems: accuracy, precision, and F1-score.

Aiming at the limitations of the ECAPA-TDNN in capturing extensive receptive fields and the problems causes by existing models in wide-range or multi-scale sound events, this paper proposes a Hierarchical-Concatenate Fusion Module (HCF) integrating the TDNN network. Meanwhile, key contributions of this study include: (1) Novel Model Design: Introducing HCF module, focusing on the effective amalgamation of multi-scale information to better capture both local and global features within Mel spectrograms to help this model get higher

**Table 1. Comparative analysis of sound event classification models.**

| Datasets | Method | Accuracy | Precision | F1-scores |
|---|---|---|---|---|
| TAU Urban Acoustic Scenes 2019 | Conv-StandardPOST | 77.2% | 78.16% | 77.7% |
| ESC-10 | Two-Stream CNN | 87.25% | 85.1% | 89.32% |
| Urbansound8k | Dual-Branch Residual Network | 82.6% | 80.7% | 84.62% |
| Urbansound8k | Net50_SE | 93.2% | 96.1% | 94.23% |
| Urbansound8k | ECAPA-TDNN | 89.0% | 84.0% | 96.0% |
| ESC-50 | CAR-Transformer | 85.72% | 95.21% | 86.67% |
| CIFAR10-AV | SMMT | 96.85% | 99.48% | 97.33% |

evaluation metrics. (2) Multi-scale Information Fusion: Parallel convolutional layers are used for multi-scale feature information fusion in HCF module to make the detection system more robust in different tasks. (3) Experimental Validation: This study contrasts HCF-TDNN with three prominent existing convolutional neural network models on the standard Urbansound8K dataset, demonstrating its state-of-the-art performance.

The GoogleNet [48] is known to have a Multiscale-multichannel feature extraction architecture compared to other pre-trained models such as ResNet. In order to overcome the limitations of sound classification models in dealing with the complexity and diversity of acoustic events, we propose a novel HCF module based on the GoogleNet for the ECAPA-TDNN network model. Although the proposed model utilizes two well-established concepts (convolution and pooling), the combination of these two concepts for Multiscale-multichannel feature extraction architecture is the first work in sound event classifications. The combination of the convolution and HCF modules is expected to work complementarily to achieve better performance in SEC. The convolution module captures the convoluted spectrum features, whereas the HCF module captures the receptive field of spectrum. To evaluate the efficacy of our method, we have conducted experiments on UrbanSound8K dataset. The results show that our method imparts the stable performance.

The rest of the paper is organized as follows: In section 2, the sound event classification model was described, including the topology of HCF-TDNN model and the details and operational principles of the HCF module structure. Section 3 details about dataset and the experimental setup. Section 4 shows results and makes an analysis in different challenge part as well. Finally, our works are concluded in section 5.

## Section 2: Sound event classification model

This section provides a detailed introduction to the model architecture and the HCF module.

### Improved model architecture design

The overall model architecture includes convolutional layer, HCF module, SE module, attention statistics pool, and fully connected layer. The detailed combination of the ECAPA-TDNN model and HCF module is shown in Fig 1.

The first layer consists of a 1-dimensional atrous convolutions layer. The main function is to gradually establish a temporal context. In addition, the first layer of convolution also improves the system's tamper-resistance performance through dimension upgrading.

The SE-HCF block is introduced into the second, third, and fourth layers. Each unit consists of two convolutional layers, an HCF module, and an SE module. This combination ensures that each unit can access all features of the input layers. For each frame layer, the

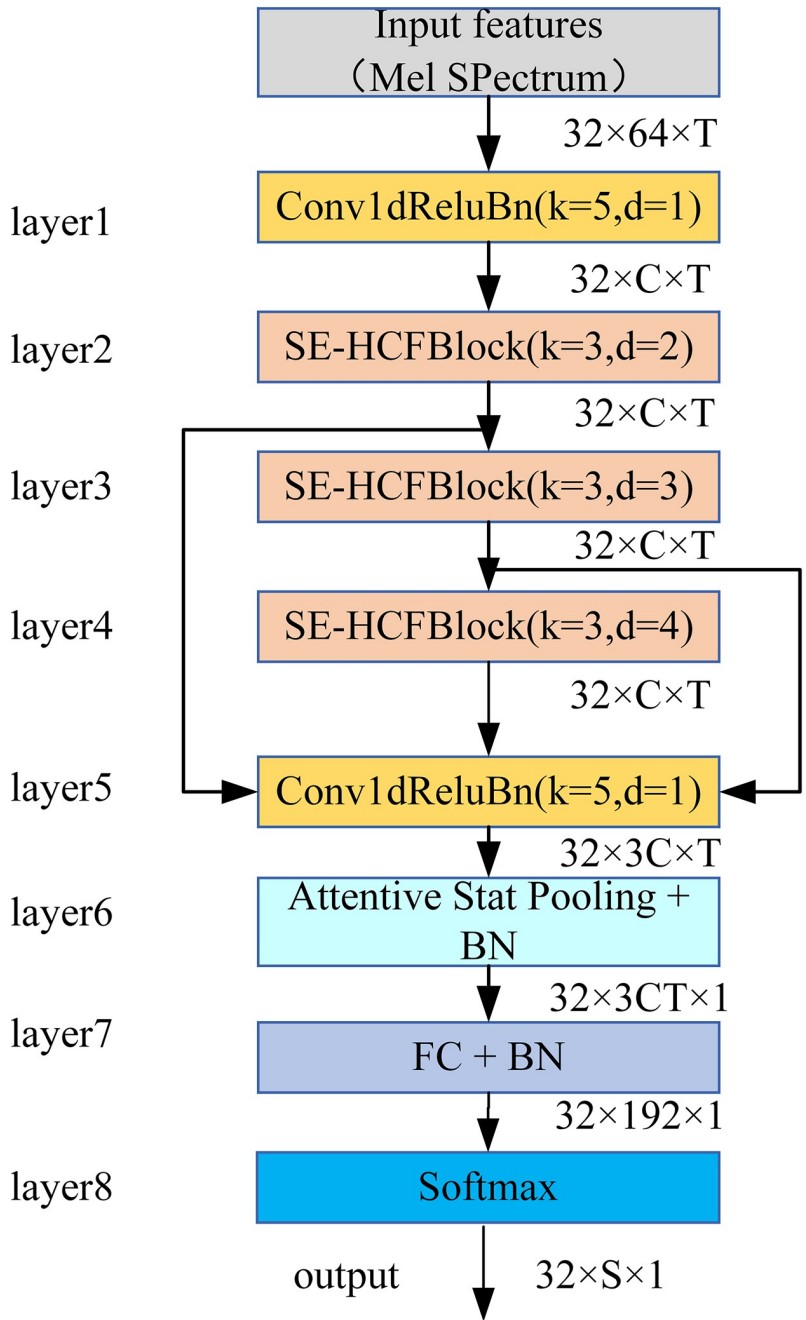

**Fig 1. A model architecture with the sound event classification model proposed by HCF-TDNN.** The HCF-TDNN branch consists of two convolution layers and three SE-HCF blocks stacked together with the Batch Normalization layer, followed by the attentive statistics pooling layer. Linear Unit (PReLU) serves as an activation function.

convolutional layers are positioned at the beginning and the end, respectively. The first convolutional layer is employed to reduce feature dimensions and computational complexity while enhancing training speed. The second convolutional layer restores the feature count to its original dimension, ensuring the integrity of feature information and facilitating feature extraction in a subsequent unit. The HCF and SE modules are in the middle of the unit. The HCF module

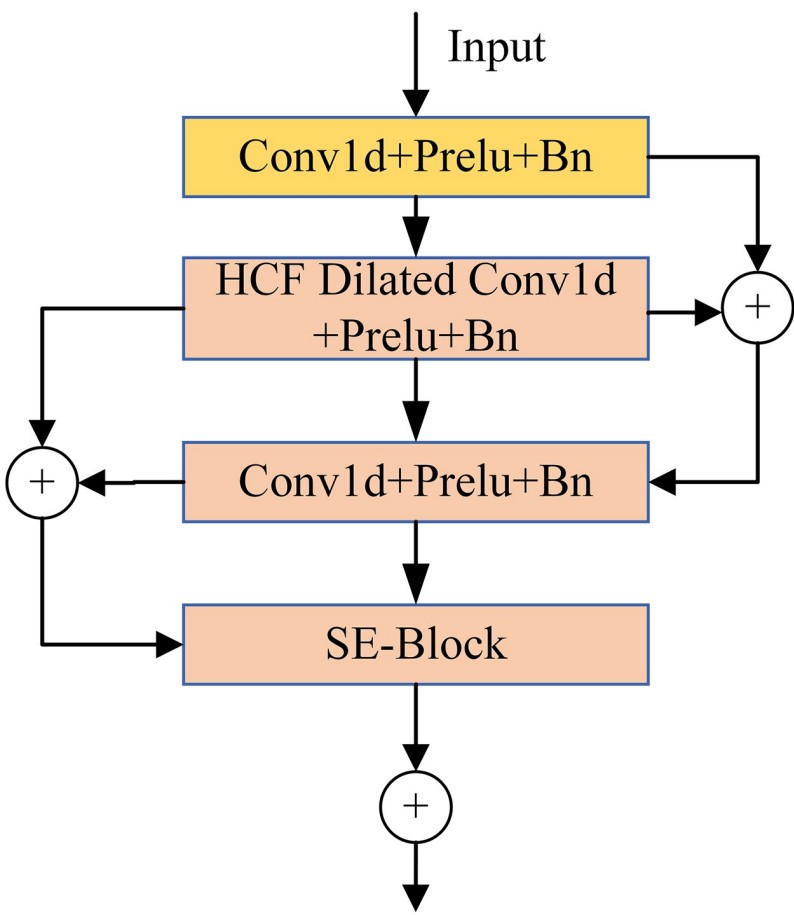

**Fig 2. The SE-HCF block of the ECAPA-TDNN architecture.** Each HCF block consists of the stack of the layers that include Conv-BN-operations and the skip-connection. The processing step contains one-dimensional convolution of the pre-trained sound features over 3-second snippets for detailed feature recognition and capturing large targets.

is utilized for detailed feature recognition and capturing large targets, while the SE module is used for channel information extraction. The entire unit employs skip connections, which is shown in Fig 2.

The fifth layer, similar to the first layer, is also a 1-dimensional atrous convolutions layer. It connects the output feature maps of all SE-HCF units to aggregate multi-layer features. Another function is to generate features for the attentive statistics pooling.

The sixth layer is the attentive statistics pooling layer. This layer assigns different weights to different frames, while generating weighted average and weighted standard deviation. Based on this way, it can effectively capture longer term changes in speaker features.

The seventh layer serves as the bottleneck layer, generating low dimensional speaker feature embedding.

The eighth layer serves as the classification layer and outputs the corresponding sound event classification results. The number of filters in the convolutional layer in the network structure is set to 128, 512, and 1024. The unbiased linear layer is used in this network, the loss function uses the Softmax with Cross-Entropy, and the epoch of the training batch is set to 50. The categories of classification are denoted as the letter S. The specific parameters of each layer in this network architecture are shown in Table 2.

**Table 2. Model parameters.**

| Sequence | Layer | Kernel | Dimension | Activation | Stride | Padding | Output |
|---|---|---|---|---|---|---|---|
| $L0$ | input | - | - | - | - | - | Batch×64×98 |
| $L1$ | Covn1d-1 | 5 | 64×512 | relu | 1 | 2 | Batch×512×98 |
| $L2$ | SE_HCF-1 | 3 | 512×512 | relu | 1 | 2 | Batch×512×98 |
| $L3$ | SE_HCF-2 | 3 | 512×512 | relu | 1 | 3 | Batch×512×98 |
| $L4$ | SE_HCF-3 | 3 | 512×512 | relu | 1 | 4 | Batch×512×98 |
| $L5$ | Covn1d-2 | 1 | 1536×1536 | relu | 1 | 2 | Batch×1536×98 |
| $L6$ | Pooling | 1 | 1536×128 | tanh | 1 | 2 | Batch×3072 |
| $L7$ | Liner-1 | - | 3072×198 | - | - | - | Batch×198 |
| $L8$ | Liner-2 | - | 198×10 | - | - | - | Batch×$S$ |

## HCF module

An improved ECAPA-TDNN module is proposed. The HCF module is inspired by GoogleNet [48], with the advantage of Plug-and-Play. Fig 3 shows the complete architecture of the HCF module.

The feature maps are divided into $s$ groups, represented by $x_1, x_2, \ldots x_s$. Each group has consistent channel width, denoted by $w$. Subsequently, each $x_i$ is processed via $s$ convolutional operations. These convolutions can be represented as $G_1, G_2, \ldots G_s$. When we perform convolutional operations on $x_1, x_2, \ldots x_s$, the output feature maps for each $x_1, x_2, \ldots x_s$ are represented as $y_1, y_2, \ldots y_s$. The innovation is an introduction to a novel concept, instead of processing these output feature maps independently, we introduce a novel concept. We concatenate adjacent feature maps $y_i$ and $y_{i+1}$ to form a subgroup, represented as $y_{i,i+1}$. Mathematically, for each $y_i$ and $y_{i+1}$, the concatenation operation is formulated as:

$$y_i = G_i(x_i) \quad i = 1 \ldots s \tag{1}$$

$$y_{i,i+1} = Concat(y_{i-1}, y_i) \quad i = 2 \ldots s \tag{2}$$

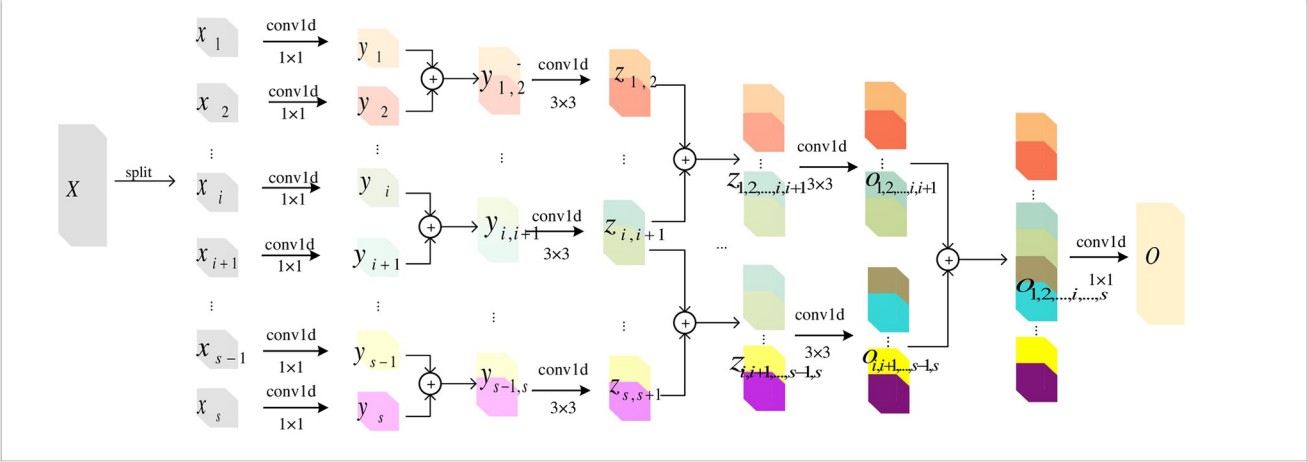

**Fig 3. A detailed view of the HCF module.** The HCF module consists of several convolutional layers. Each layer contains a unique set of filters for extracting temporal features from the input audio signals. These filters are crafted to respond to different frequency ranges and time scales, allowing the module to capture a wide range of temporal characteristics in sound events.

**Table 3. HCF module parameters.**

| Sequence | Layer | Kernel | Dimension | Activation | Stride | Padding | Output |
|---|---|---|---|---|---|---|---|
| $L0$ | input | - | - | - | - | - | Batch×512×98 |
| $L1$ | Covn1d-1 | 3 | 128×128 | relu | 1 | 2 | Batch×128×98 |
| $L2$ | Covn1d-2 | 3 | 256×256 | relu | 1 | 2 | Batch×256×98 |
| $L3$ | Covn1d-3 | 3 | 512×512 | relu | 1 | 2 | Batch×512×98 |
| $L4$ | Covn1d-4 | 3 | 1024×1024 | relu | 1 | 2 | Batch×1024×98 |
| $L5$ | Covn1d-5 | 3 | 1024×512 | relu | 1 | 2 | Batch×512×98 |

Where, "Concat" denotes the concatenation of two feature maps along the channel dimensions. After pairwise concatenation, $s - 1$ subgroups are formed. Then, these subgroups are fed into a subsequent series of layers for further convolution and adjacent feature map concatenation until forming a single subgroup. Similarly, in Fig 3, $z_{i,i+1}$, $o_{1,2,\dots,i,i+1}$, and $o_{i,i+1,\dots,s-1,s}$ represent the results of subgroups convolution operations. After operating all input feature groups, $o_{1,2,\dots,i,i+1,\dots,s-1,s}$ restores the channel dimension by concatenating all and then utilizing 1D convolution to reduce the dimensionality. The result after dimensionality reduction is denoted by $O$.

The method of creating subgroups by concatenating adjacent feature maps within each group enhances the model's ability to capture both local and contextual information. It provides more structured features, which can improve performance in sound classification by CNN.

Within this module, parameters and computational complexity are reduced by controlling the number of convolution groups, layers, and channels. The larger the number of groups and layers, the stronger the multi-scale extraction capability. The more the channels are, the richer features become. The specific parameters of the proposed HCF module are shown in Table 3.

## Split and concatenate operation of HCF module

The HCF module consists of two primary stages: the split and concatenate operations.

- Splitting Operation: Initially, the integrated feature map $X$ is divided into different groups, denoted by $x_i$. Importantly, each of these groups $x_i$ maintains the same width.

- Phase of Connection Operation: This operation combines adjacent feature maps $y_i$ and $y_{i+1}$ to form a subgroup $y_{i,i+1}$. It's worth noting that the channel width of $y_{i,i+1}$ is twice $y_i$ or $y_{i+1}$.

- Purpose of Identity Mapping: The identity mapping for $y_i$ aims to preserve the intrinsic feature maps. Conversely, $y_{i,i+1}$ focus on capturing more complex features.

- Channel Expansion & Information Flow: Each connection (such as the initial connection between $y_i$ and $y_{i+1}$, etc.) is utilized to widen the channel width. This amplifies the exchange of information between different groups, enhancing feature representation.

- Final Connection & Feature Maintenance: All subgroups are connected, and their outputs are processed by a 1×1 convolutional layer. In summary, the HCF module is employed a combination of splitting and connecting operations. This design allows for effective channel expansion, facilitates the enhancement of information flow, and maintains the integrity of feature representation, distinguishing it from summation methods used in Res2Net [31].

## Section 3: Experimental setup

In this section, we primarily describe the experiment conducted to evaluate the proposed acoustic scene classification. Two contrasting methods, vertical contrast and horizontal contrast, are employed to ensure the experiment validity. The vertical experiment mainly focus on comparing baseline models ECAPA-TDNN [39], AF-TDNN [40], ECAPA-TDNN-GRU [49] and MultiChannel-ECAPA-TDNN [50]. Horizontal experiment primarily compare with the conformer [46], VGG19 [51] and STF-Yolo8 [52]. These model comparisons are implemented in Pycharm with the Torchaudio and PyTorch libraries [53]. In this experiment, training is accomplished using a GPU (NVIDIA GeForce GTX 3060).

### Datasets

The public dataset Urbansound8k is adopted in this experiment, which contains 8732 labelled excepts of urban sounds from 10 categories (durations less than or equal to 4s, a total of 9.7h) [33]. Urbansound8k is a medium-sized audio dataset. The length of each segment is at least 3 seconds. Considering the amount of Urbansound8k data, the training and testing data are divided into 7032 and 1700 pieces in independent experiment, respectively. The test dataset is extracted for 170 audio data from each category.

### Acoustic features

In the experiment, each input audio is converted into a 64 dimensional Mel spectrograms through a short-term Fourier transform (STFT) and subsequent Mel filter bank. The Mel spectrograms are generated by STFT with 25ms Hamming windows duration and 10ms shift. Mean normalization is used for each frequency interval of the spectrum.

### Network settings

The proposed model is trained on 50 epochs and with a batch size of 32. To optimize the network, the Adam optimizer is carried out with a learning rate of 0.001. The unbiased linear layer is used in this network. When verifying the sound events, Softmax activation is usually adopted at the output layer. For continuous control of the training and prevention of overfitting, the experimental process involved one round of training and one round of evaluation.

### Evaluation metrics

For the sake of monitoring the performance of the models, Accuracy, Kappa, and F1 score are used for experimental method evaluation. In the statistical results, there are four relationships between actual and predicted values. TP (True Positive) indicates that observation is positive, and prediction is positive; FP (False Positive) indicates that observation is negative, but prediction is positive; FN (False Negative) indicates that observation is positive, but prediction is negative; TN (True Negative) indicates that observation is negative, and prediction is negative. The Accuracy, Kappa, and F1-scores are defined as:

$$Accuracy = \frac{TP + TN}{TP + FP + TN + FN} \tag{3}$$

$$Precision = \frac{TP}{TP + FP} \tag{4}$$

$$Recall = \frac{TP}{TP + FN} \tag{5}$$

$$F1 = \frac{2}{\frac{1}{Precision} + \frac{1}{Recall}} \tag{6}$$

$$Ex\_Acc = \frac{(TP + FN) \times (TP + FP) + (FP + TN) \times (FN + TN)}{(TP + FP + TN + FN)^2} \tag{7}$$

$$Kappa = \frac{Accuracy - Ex\_Acc}{1 - Ex\_Acc} \tag{8}$$

## Section 4: Experimental results

To further demonstrate that the proposed HCF module can effectively enhance the sound classification accuracy of ECAPA-TDNN, this section will analyze and discuss the performance of HCF module by ablation experiment and data analysis.

### Ablation experiment

To ascertain the improvement of different activation functions for the proposed model in this paper, classification combinations of the model are tested. In the same model backbone network, tests are conducted using three different activation functions: Relu, Leaky ReLU with a negative slope of 0.02, and PReLU (Parametric Rectified Linear Unit) with a learning parameter of 0.1. The results of these tests are shown in Table 4.

In the experiment, compared to Relu, Leaky Relu can improve the model's accuracy by about 0.35%. PReLU further enhances the accuracy by approximately 1.1%. Therefore, the PReLU activation function is employed for experiment in the HCF-TDNN model.

To further analyze the impact of different blocks within the HCF module on the overall system, ablation experiments are executed on the HCF module. The specific experimental results are shown in Table 5.

**Table 4. The model accuracy and F1 score under different activation functions.**

| Activation Function | Method | Accuracy | F1-scores |
|---|---|---|---|
| *Relu* | HCF-TDNN | 94.44% | 95.92 |
| *LeakyReLU* | HCF-TDNN | 94.79% | 95.88 |
| *PReLU* | HCF-TDNN | 95.83% | 97.04 |

**Table 5. Ablation study of the HCF-TDNN architecture.**

| Exp | SE | HCF | Skip Connections | Accuracy | Precision | Sensitivity | Recall | F1-scores |
|---|---|---|---|---|---|---|---|---|
| *Exp*.1 | × | × | × | 84.72% | 85.14% | 98.36% | 83.75% | 84.44 |
| *Exp*.2 | × | × | ✓ | 85.42% | 85.77% | 98.43% | 87.68% | 86.71 |
| *Exp*.3 | × | ✓ | × | 88.54% | 87.73% | 98.8% | 92.5% | 90.58 |
| *Exp*.4 | ✓ | × | × | 87.85% | 89.28% | 98.7% | 90.39% | 89.83 |
| *Exp*.5 | × | ✓ | ✓ | 89.24% | 90.48% | 98.8% | 92.5% | 91.48 |
| *Exp*.6 | ✓ | × | ✓ | 89.58% | 89.98% | 98.81% | 91.96% | 91.67 |
| *Exp*.7 | ✓ | ✓ | × | 93.40% | 94.16% | 99.34% | 96.04% | 95.09 |
| *Exp*.8 | ✓ | ✓ | ✓ | 95.83% | 96.59% | 99.6% | 97.5% | 97.04 |

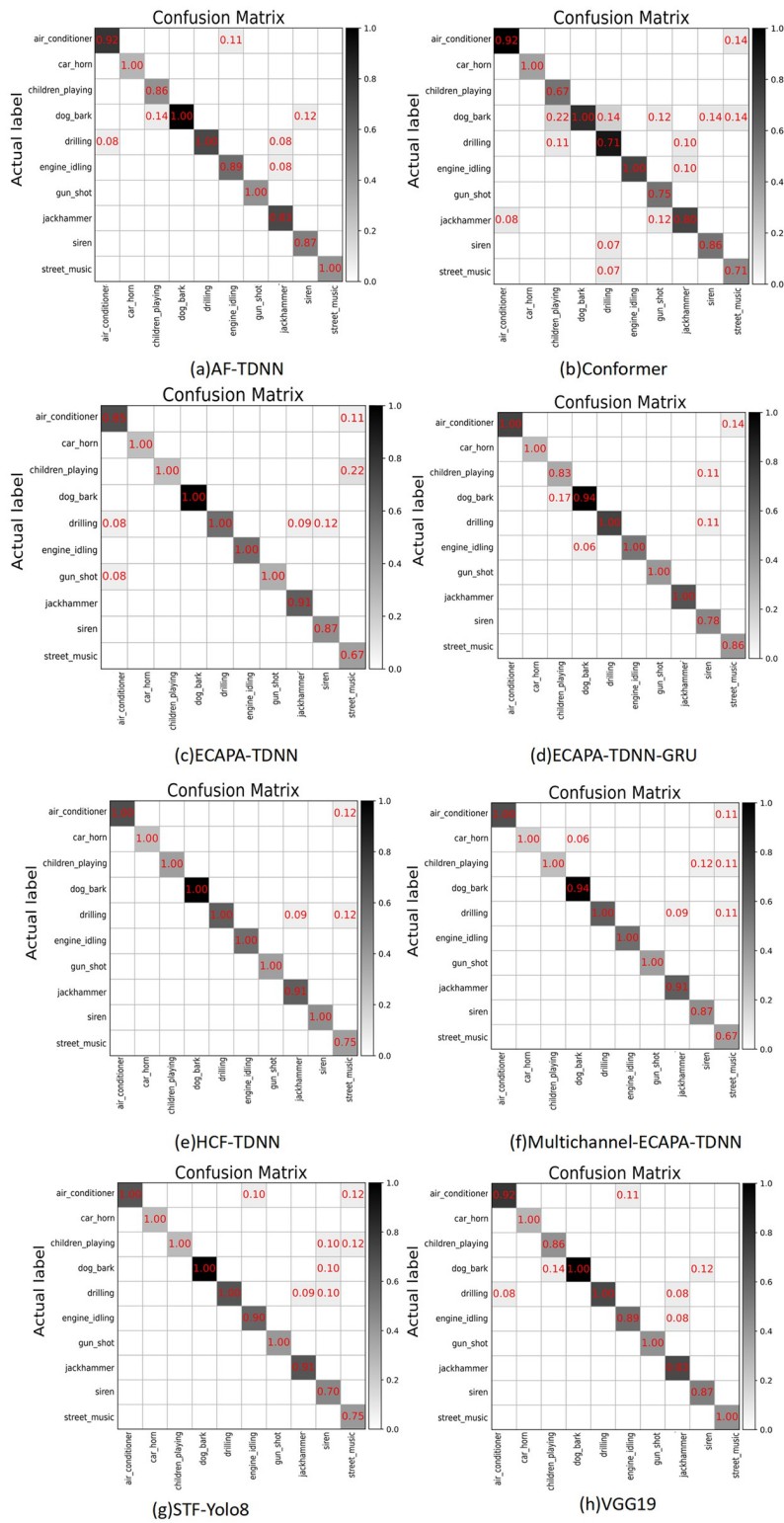

**Fig 4. Comparison of confusion matrices for eight models in sound classification.** The performance of sound category classification is meticulously detailed portrayals by each matrix. The matrices are clearly organized for easy comparison of TP, FP, TN, and FN across models. The strengths and weaknesses of models in sound type differentiation are indicated by the color gradients in each matrix.

The experimental results indicate that the sound classification accuracy is lower than that of the baseline model with the SE module, HCF module, or skip connection individually. When combining the SE module with the HCF module, the accuracy is 93.04%, representing an improvement of approximately 3%. The simultaneous addition of all three components increases the accuracy to 95.83%, achieving the highest classification precision.

## Comparative analysis

We improve the ECAPA-TDNN model by replacing Res2net with the HCF module proposed in this paper. The confusion matrices for each model are shown in Fig 4.

By comparing the probability values on the diagonals of each model, the HCF-TDNN achieves a prediction accuracy of 100% for eight types of sounds, including air conditioner and car horn, surpassing the overall sound recognition performance of other models.

To further evaluate the performance of the four different models in sound event classification, a box plot is generated to visualize the distribution of F1 scores for each model, as shown in Fig 5.

Through the analysis of the box plot results for F1 scores, we can clearly compare and interpret the performance of various models in audio classification. It can be seen from median F1 scores that the HCF-TDNN and ECAPA-TDNN exhibit similar performance. Therefore, it is necessary to further assess the models comprehensively using Kappa values, providing a more

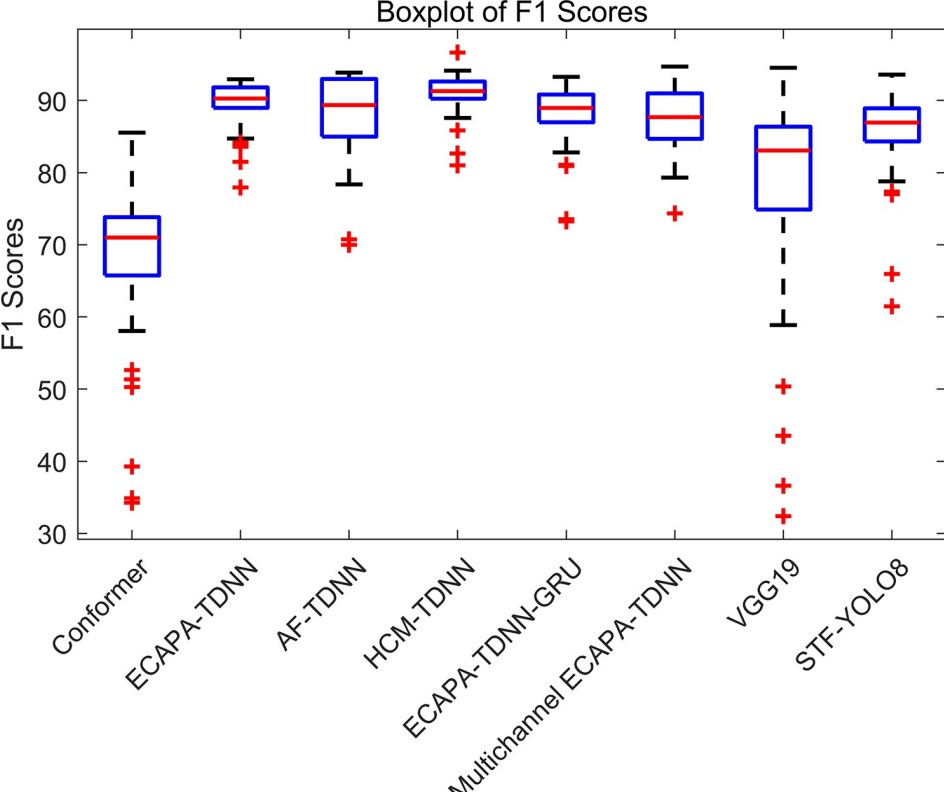

**Fig 5. Box plot analysis of F1 scores for various models.** In SEC, the F1 score range and potential outliers obtained by various models are vividly outlined by the block diagram for each model. The effectiveness of different sound scapes in complex tasks is distinguished by the visualization of box plots.

holistic evaluation than simple accuracy, especially in cases of class imbalance or uneven errors, as shown in Fig 6.

It should be noted that, for the median Kappa values, the HCF-TDNN model demonstrates the highest predictive consistency. Additionally, the HCF-TDNN model exhibits relatively short whiskers, indicating stable performance.

For better visual observation, the accuracy results for each neural network model are plotted as curves. The loss curves and test accuracy curves for each model are shown in Figs 7 and 8, respectively. Since the same number of sound categories are used to form the test set in this study, the influence of randomness related to initializing weights and hidden layer units can be disregarded.

Clearly, with the increase of epochs, the loss curve of the HCF-TDNN model gradually decreases to a relatively stable value. Additionally, the accuracy of the HCF-TDNN model gradually improves, indicating that the model has reached its optimal performance.

Compared to the Transformer model, the HCF-TDNN model not only enhances the recognition performance but also demonstrates strong convergence. In terms of the same experimental conditions, the accuracy of the HCF-TDNN model is approximately 3%-5% higher than that of AF-TDNN and ECAPA-TDNN. The experimental results for ECAPA-TDNN,

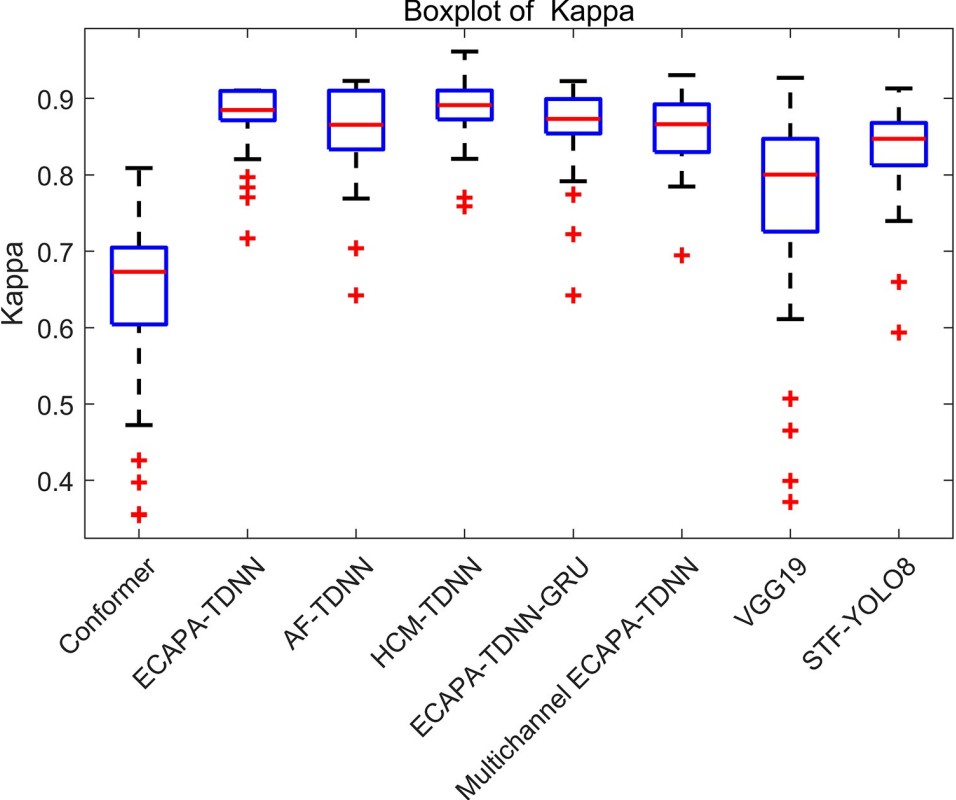

**Fig 6. Comparison box plot of Kappa values for different models in sound classification.** The Kappa coefficients of multiple models within the sound classification domain are evaluated through the presented box plot. The distribution and consistency of Kappa scores are summarized graphically. The degree of agreement between actual and predicted sound classifications is measured by the box plot. The identification of models achieving higher precision and consistency in differentiating sound types is facilitated by the Kappa scores.

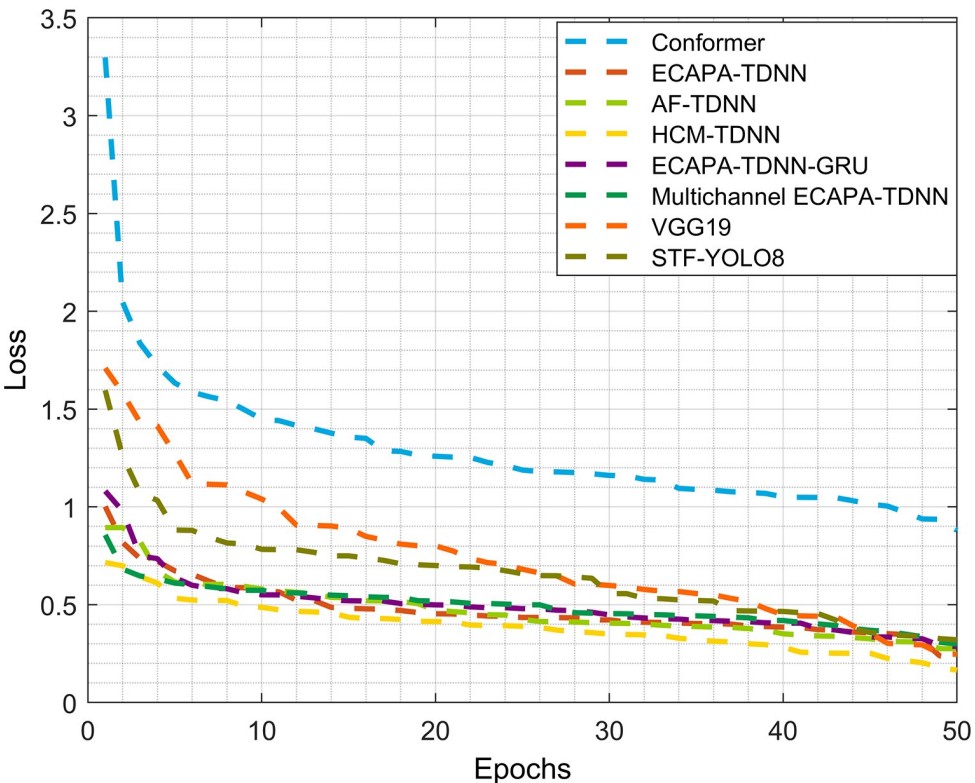

**Fig 7. Comparison of Loss curves across various models on UrbanSound8K dataset.** The evolutions of each model's performance over different epochs are illustrated by the graph. The learning dynamics and convergence rates of various models are revealed by this visualization. A visual assessment of model convergence and training effectiveness is offered by the distinct learning curves. The models that struggle to minimize the loss are also identified by these curves.

AF-TDNN, Transformer, and HCF-TDNN model in sound event classification are presented in Table 6.

The results indicate that the testing performance of the HCF-TDNN model is superior to other models. This is attributed to the proposed model in this study, which not only considers how to extract multi-scale features to enhance feature aggregation but also places emphasis on increasing the channel count and enriching channel information by adding an appropriate number of layers.

## Section 5: Conclusion

This paper proposes a novel HCF module based on the ECAPA-TDNN model to improve model performance. The HCF-TDNN model uses Mel spectrograms as input features. It is possible to capture details while still considering local features by the idea of combining adjacent features. The combination of features from different levels can better integrate semantic information. Finally, HCF-TDNN model is compared with the previous work in the same experimental environment. Based on the experimental results, HCF-TDNN model can achieve an average recognition accuracy of 95.83% on the dataset UrbanSound8K. On the prospects of systems design, the optimizations targeted at embedded deployment can help realize real-time urban environmental sound recognition scenarios and edge

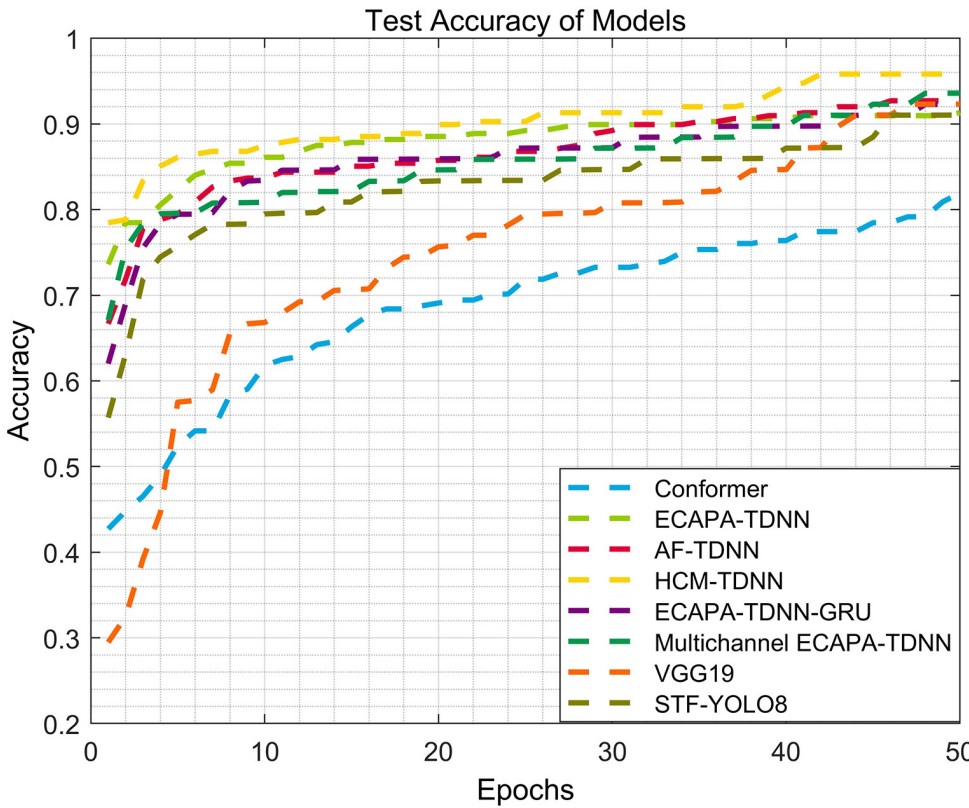

**Fig 8. Model testing accuracy curves based on UrbanSound8K dataset.** Evolutions of the loss function over the epochs during the training using the training and the validation datasets for the different cases are evaluated. The testing accuracy curves also reveal the model's performance on SEC.

computing solutions. This research makes notable progress in sound event detection, and the presented approach, analyses, and directions can be laying the groundwork to enable smart city design. Future work should focus on further compressing the model design as well as incorporating additional shape and context information and exploring supplementary data sources.

**Table 6. Comparison of datasets.**

| Datasets | Method | Accuracy | Precision | Sensitivity | Recall | F1-scores | Kappa |
|---|---|---|---|---|---|---|---|
| UrbanSound8K | Conformer | 81.944% | 84.19% | 98.0% | 86.92% | 85.53% | 0.8089 |
| | ECAPA-TDNN | 90.625% | 91.33% | 92.44% | 93.75% | 92.52% | 0.9103 |
| | AF-TDNN | 92.708% | 93.0% | 99.22% | 94.0% | 93.5% | 0.923 |
| | **HCF-TDNN** | **95.83%** | **96.59%** | **99.6%** | **97.5%** | **97.04%** | **0.9615** |
| | ECAPA-TDNN-GRU | 93.056% | 94.06% | 99.35% | 94.93% | 94.49% | 0.936 |
| | Multichannel ECAPA-TDNN | 92.014% | 94.33% | 99.24% | 92.22% | 93.26% | 0.9227 |
| | VGG19 | 92.708% | 93.71% | 99.2% | 95.14% | 94.42% | 0.9231 |
| | STF-Yolo8 | 91.319% | 90.89% | 99.09% | 92.08% | 91.48% | 0.9103 |

## Author Contributions

**Investigation:** Jiwen Liang.

**Methodology:** Jiwen Liang.

**Supervision:** Baishan Zhao.

**Validation:** Baishan Zhao.

**Visualization:** Jiwen Liang.

**Writing – original draft:** Baishan Zhao, Jiwen Liang.

**Writing – review & editing:** Baishan Zhao.

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
