## [Decision Letter · Decision Letter 0]

27 Dec 2023

PONE-D-23-39866Hierarchical-Concatenate Fusion TDNN for Sound Event ClassificationPLOS ONE

Dear Dr. Liang,

Thank you for submitting your manuscript to PLOS ONE. After careful consideration, we feel that it has merit but does not fully meet PLOS ONE’s publication criteria as it currently stands. Therefore, we invite you to submit a revised version of the manuscript that addresses the points raised during the review process.

We look forward to receiving your revised manuscript.

Kind regards,

Ali Haider Khan

Academic Editor

PLOS ONE

Journal Requirements:

Additional Editor Comments:

The manuscript is interesting and well organized, with detail and relevant information. However, it needs to be revised as per the details:

The research problem article is addressing needs to be clearer and must be defined in the introduction section clearly.

The research objectives are not defined clearly in the introduction section.

There is a serious concern, authors should conduct a comparative analysis to prove that their research has better performance, and addressing all the objectives of this proposed system.

Methodology relating to the objectives of this research is not satisfactory, need revision to fulfill all the objectives of this research.

Reviewers' comments:

Reviewer's Responses to Questions

**Comments to the Author**

1. Is the manuscript technically sound, and do the data support the conclusions?

Reviewer #1: Yes

Reviewer #2: Partly

2. Has the statistical analysis been performed appropriately and rigorously? 

Reviewer #1: Yes

Reviewer #2: No

3. Have the authors made all data underlying the findings in their manuscript fully available?

Reviewer #1: Yes

Reviewer #2: Yes

4. Is the manuscript presented in an intelligible fashion and written in standard English?

Reviewer #1: Yes

Reviewer #2: Yes

5. Review Comments to the Author

Reviewer #1: There are certain points that need to be addressed:

1: The author claims that "to solve the above problem, HCM is proposed," but the author did not specify the above problem.

2: The authors did not follow English rules. There should be some space between two sentences, e.g., (I went to shop. I bought eggs.)

2.1: All the text should be aligned, which does not seem to be the case in the research.

3: The authors did not mention their contributions specifically in the current research.

4: Overall, this research sounds good.

5: Recommended with minor changes.

Reviewer #2: In figure 4, except conformer, all the methods showing and achieved same accuracy till epoch 29, justify it?

is the code for this experiment is publicly available for authenticity?

why you use the technique Mel spectrograms as input features in this proposed network?

Have you check the accuracy of your proposed techniques on lesser or more epoch(i.e at 20-25 or at 35-40) ?

6. PLOS authors have the option to publish the peer review history of their article (what does this mean?). If published, this will include your full peer review and any attached files.

Reviewer #1: **Yes: **Muhammad Sajid

Reviewer #2: **Yes: **Tahir Abbas

---

## [Author Response · Author response to Decision Letter 0]

11 Feb 2024

Dear Ali Haider Khan,

I would like to express my sincere gratitude to you and the editorial team of Plos One for the meticulous review of our SCI paper and for providing valuable feedback. After careful revisions, we have completed the necessary changes and are prepared to submit the final version of the manuscript.

Here is our detailed response to each of the reviewers' comments:

[Response to Reviewers] file: We have attached a document responding to each of the reviewers' comments, outlining specific modifications and providing explanations for our choices.

[Revised Manuscript with Track Changes] file: We are submitting a revised manuscript with tracked changes, highlighting the modifications made in response to the reviewers' suggestions.

[Manuscript] file: Additionally, we have included an unmarked version of the revised paper for your convenience in reviewing the final version of the manuscript.

We greatly appreciate your guidance and the reviewers' insightful comments throughout the review process. The constructive feedback has not only enhanced the quality of our paper but also provided profound insights into our research.

We will submit the revised manuscript as per your specified instructions, ensuring that it aligns with the publication standards of [Journal Name]. If you require any additional information or have further matters that need our attention, please feel free to inform us.

Thank you once again for your professional guidance and your patience in awaiting our revised submission. We look forward to receiving the final approval and hope that the paper meets your expectations for publication in [Journal Name].

Thank you for your time and effort.

Best regards,

Jiwen Liang, Ph.D. School of Information Science and Engineering, Shenyang University of Technology

111 Shenliao West Road, Economic and Technological Development, Shenyang, Liaoning

Province, China

Telephone Number: +8618624341326

E-mail Addresses: a17741226289@163.com

---

## [Decision Letter · Decision Letter 1]

21 Mar 2024

PONE-D-23-39866R1Hierarchical-Concatenate Fusion TDNN for Sound Event ClassificationPLOS ONE

Dear Dr. Liang,

Thank you for submitting your manuscript to PLOS ONE. After careful consideration, we feel that it has merit but does not fully meet PLOS ONE’s publication criteria as it currently stands. Therefore, we invite you to submit a revised version of the manuscript that addresses the points raised during the review process.

We look forward to receiving your revised manuscript.

Kind regards,

Ali Haider Khan

Academic Editor

PLOS ONE

Reviewers' comments:

Reviewer's Responses to Questions

**Comments to the Author**

1. If the authors have adequately addressed your comments raised in a previous round of review and you feel that this manuscript is now acceptable for publication, you may indicate that here to bypass the “Comments to the Author” section, enter your conflict of interest statement in the “Confidential to Editor” section, and submit your "Accept" recommendation.

Reviewer #1: All comments have been addressed

Reviewer #3: All comments have been addressed

Reviewer #4: (No Response)

2. Is the manuscript technically sound, and do the data support the conclusions?

Reviewer #1: Yes

Reviewer #3: Yes

Reviewer #4: (No Response)

3. Has the statistical analysis been performed appropriately and rigorously? 

Reviewer #1: Yes

Reviewer #3: Yes

Reviewer #4: (No Response)

4. Have the authors made all data underlying the findings in their manuscript fully available?

Reviewer #1: Yes

Reviewer #3: Yes

Reviewer #4: (No Response)

5. Is the manuscript presented in an intelligible fashion and written in standard English?

Reviewer #1: Yes

Reviewer #3: Yes

Reviewer #4: No

6. Review Comments to the Author

Reviewer #1: All the comments have been corrected, and the article is now written in proper shape.

Recommended to accept

Reviewer #3: (No Response)

Reviewer #4: Review Comments

(1) The abstract is poorly written. Most of the content focuses on the outcome of the proposed model. It should include a little bit of problem background and research objectives as well.

(2) The Introduction section should provide a brief introduction for the rest of the article's contents instead of the literature review-related contents.

(3) The literature review should be presented comparatively to show the research gap quantitatively (if possible). For instance, the author can identify some parameters and compare the related models/methodologies based on them in a tabular form.

(4) The research objectives, aims, motivations, and ultimate beneficiaries should be well explained to show the significance of the research contribution.

(5) The graphics quality of figures should be improved so that the textual information should be better readable.

(6) It is suggested to add a little bit more description after each figure instead of giving their labels only. For reference, please see figures from numbers 4 to 7.

(7) The two headings are being repeated. See line number 50 and 181.

(8) Future directions should also be discussed to explain the possible future advancements in the underlined research area.

(9) The referenced articles are fewer in number to better support the importance and significance of the underlying study. It is recommended to increase the number of referenced articles by adding more related models/methodologies to be compared with the proposed one.

10) Some spelling and grammatical mistakes are found in the article that must be corrected. For example see spelling of “classifiy”, “labeled”, “analysises”, etc., and a grammatical mistake like “…achieving an low Equal Error”, “…combined in pair”, “…are formed. And then, 101 these sub-groups are”

7. PLOS authors have the option to publish the peer review history of their article (what does this mean?). If published, this will include your full peer review and any attached files.

Reviewer #1: **Yes: **Muhammad Sajid

Reviewer #3: **Yes: **Tahir Abbas

Reviewer #4: No

---

## [Author Response · Author response to Decision Letter 1]

4 May 2024

Comments from Reviewer：

Reviewer #1:

(1)Comment: All the comments have been corrected, and the article is now written in proper shape.Recommended to accept.

Author Response: 

Thank you for your positive feedback on our manuscript titled "Hierarchical-Concatenate Fusion TDNN for Sound Event Classification." 

Reviewer #2:

(1)Comment: The abstract is poorly written. Most of the content focuses on the outcome of the proposed model. It should include a little bit of problem background and research objectives as well.

Author Response:

Thank you for your constructive feedback regarding the abstract.

We acknowledge the shortcomings of the abstract. We have re-written this part according to the reviewer. Specific modifications are as follows:

First, we have corrected the "Semantic feature combination/parsing issue is one of the key problems in sound event classification." into "Semantic feature combination/parsing issue is one of the key problems in sound event classification for acoustic scene analysis, environmental sound monitoring, and urban soundscape analysis."

Second, we have corrected the"Hierarchical-Concatenate Model (HCM) is proposed to improve the performance of ECAPA-TDNN Networks." into "TThis paper proposes the Hierarchical-Concatenate Fusion(HCF)-TDNN model by adding HCF Module to ECAPA-TDNN model for sound event classifica- tion."

 Third, we have corrected the"Second, the segmented features are convolved one by one. Finally, after the convolution is completed, the two adjacent parts are combined before proceeding with the next convolution." into "Then, the segmented features are convolved one by one for improving the small receptive field in perceiving details. Finally, after the convolution is completed, the two adjacent parts are combined before proceeding with the next convolution for enlarging the receptive field in capturing large targets."

 Finally, we have added the"The proposed model is trained and validated on the Urbansound8K dataset." and " which is approximate improvement of 5% (relatively) over ECAPA-TDNN model." (please see: Abstract ).

Once again, we sincerely appreciate your valuable feedback and insights, which significantly contribute to refining our research.

(2)Comment: The Introduction section should provide a brief introduction for the rest of the article's contents instead of the literature review-related contents.

Author Response:

Thank you for your valuable feedback regarding the Introduction section.

Yes, We acknowledge the need to refocus the Introduction to provide a brief overview for the rest of the article's contents. In the introduction section, the rest of the article's contents have been added. Specifically:

"The rest of the paper is organized as follows: In section 2, the sound event classification model was described, including the topology of HCM-TDNN and the details and operational principles of the HCM module structure. Section 3 details about dataset and the experimental setup. Section 4 shows results and makes analysis in different challenge part as well. Finally, our works are concluded in section 5".(please see:p.2 lines 44-48).

Thank you once again for your valuable feedback and for emphasizing the importance of authenticity in scientific endeavors. 

(3)Comment: The literature review should be presented comparatively to show the research gap quantitatively (if possible). For instance, the author can identify some parameters and compare the related models/methodologies based on them in a tabular form.

Author Response:

Thank you for your suggestion regarding the presentation of the literature review.

We acknowledge the importance of presenting the literature review in a comparative manner to highlight quantitative research gaps. We have enhanced our manuscript by comparing related models, as suggested. Specifically:

We have corrected the "Notably, the ResNet combined with the SE module [11], Convolutional Neural Networks (CNN) [12][13][14], and approaches grounded in Transformer Networks have been explored. " into " Notably, the ResNet[13] combined with the SE module [14] to enhance the robustness of classification model. Convolutional Neural Networks (CNN) [15-17] have become the standard method for addressing various audio classification problems. And approaches grounded in Transformer Networks have achieved great success in the text recognition task."(please see:p.2 lines 13-18).

Besides, we have revised the " Furthermore, prevalent neural network architectures are based on Time Delay Neural Networks (TDNN) [16], also known as x-vector architectures." into "Furthermore, prevalent neural network architectures are based on Time Delay Neural Networks (TDNN) [19], which widely used in Automatic Speaker Verification, such as x-vector system." (please see:p.2 lines 19-21).

(4)Comment: The research objectives, aims, motivations, and ultimate beneficiaries should be well explained to show the significance of the research contribution.

Author Response:

Thank you for your valuable feedback regarding the clarity of the research objectives, aims, motivations, and ultimate beneficiaries.

We have carefully read all suggestions by the reviewer and added corresponding descriptions of the research objectives, aims, motivations, and ultimate beneficiaries in the introduction section on page 2.

We have revised the"Key contributions of this study include: (1) Novel Network Design: Proposing HCM, focusing on the effective fusion of multi-scale information to better capture both local and global features within Mel spectrograms. (2) Multi-scale Information Fusion: Through parallel convolutional layers in HCM, the model adeptly manages features of varying scales, significantly boosting the recognition of micro & macro features within Mel spectrograms."into"Meanwhile, key contributions of this study include: (1) Novel Model Design: Introducing HCF module, focusing on the effective amalgamation of multi-scale information to better capture both local and global features within Mel spectrograms to help this model get higher evaluation metrics. (2) Multi-scale Information Fusion: Parallel convolutional layers are used for multi-scale feature information fusion in HCF module to make the detection system more robust in different tasks".(please see:p.2 lines 36-41).

We sincerely value your expertise, which has played a crucial role in shaping the final version of our manuscript. Your thorough review has been indispensable in elevating the quality of our research.

(5)Comment: The graphics quality of figures should be improved so that the textual information should be better readable.

Author Response:

Thank you for your feedback regarding the graphics quality of the figures.

We acknowledge the importance of enhancing the clarity and readability of the textual information within the figures. We have improved the quality of full-text images by increasing the image resolution, using vector graphics, redrawing images and other methods.

In this updated version, we improved figures 1-8 in the main text. Details can be found in (Fig 1, page 3; Fig 2, page 3; Fig3, page4; Fig4, page9; Fig5, page10; Fig6, page10; Fig7, page11; Fig 8, page 11).

"We have changed the captions of Fig 4, Fig 5, Fig 6 and Fig 7 from " Confusion matrix for (a) Conformer, (b) AF-TDNN, (c)ECAPA-TDNN and(d)HCM-TDNN." to " Comparison of Confusion Matrices for Eight Models in Sound Classification." (please see: p.9 ), " Comparison Box Plot of F1 Scores." to " Box Plot Analysis of F1 Scores for Various Models." (please see:p.10 ), " Comparison Box Plot of Kappa." to " Comparison Box Plot of Kappa Values for Different Models in Sound Classification." (please see:p.10 ) and " Loss curves of models." to " Comparison of Loss Curves Across Various Models on UrbanSound8K Dataset." (please see:p.11).

(7)Comment: The two headings are being repeated. See line number 50 and 181.

Author Response:

Thank you for pointing out the repetition of the two headings. We have revised the document to ensure that each heading is unique and that there are no repetitions throughout the text. Specifically：

The heading "SOUND EVENT CLASSIFICATION MODEL" has remained unchanged. (please see:p.2 lines 49). And the heading "SOUND EVENT CLASSIFICATION MODEL" has been changed to "EXPERIMENTAL RESULTS". (please see:p.7 lines 181).

(8)Comment: Future directions should also be discussed to explain the possible future advancements in the underlined research area.

Author Response:

Thank you for your suggestion. We agree that discussing future directions is important to explain the possible future advancements in the research area. 

We have supplemented "In future work, the voice recognition system HCM-TDNN model will be deployed in edge computing devices for real-time urban environmental sound recognition scenarios." (please see:p.12 lines 248-250).

(9)Comment: The referenced articles are fewer in number to better support the importance and significance of the underlying study. It is recommended to increase the number of referenced articles by adding more related models/methodologies to be compared with the proposed one.

Author Response:

Thank you for your valuable feedback. We recognize the need to increase the number of referenced articles to better support the importance and significance of our study. 

In the revised manuscript, We have added more relevant models and expanded the list of cited articles. The number of citations has increased by 8 compared to the first revised manuscript, bringing the overall number to 32. Particularly:

[1]Mashhadi MMR, Osei-Bonsu K. Speech Emotion Recognition Using Machine Learning Techniques: Feature Extraction and Comparison of Convolutional Neural Network and Random Forest. PLOS ONE. 2023 Nov;18(11):e0291500.

[2]Yue Z, Loweimi E, Christensen H, Barker J, Cvetkovic Z. Acoustic Modelling from Raw Source and Filter Components for Dysarthric Speech Recognition. IEEE/ACM Trans AudioSpeech and Lang Proc. 2022 Sep;30(1):2968-2980.

[3]Du XW, Si LQ, Li PF, Yun ZH. A Method for Detecting the Quality of Cotton Seeds Based on An Improved Resnet50 Model. PLOS ONE. 2023 Feb;18(2):e0273057.

[4]Diaz-Escobar J, Ord´o˜nez-Guill´en NE, Villarreal-Reyes S, Galaviz-Mosqueda A, Kober V, Rivera-Rodriguez R, Rizk JEL. Deep-Learning Based Detection of Covid-19 Using Lung Ultrasound Imagery. PLOS ONE. 2021;16(8):e0255886.

[5]Haitao C, Yu L, Yun Y. Research on Voiceprint Recognition System Based on ECAPA-TDNN-GRU Architecture. 2023 IEEE 2nd International Conference on Electrical Engineering, Big Data and Algorithms (EEBDA); 2023.

[6]Xin A, Haitao Z, Shuai Z. ASC Model Based on Feature Stratification and Multichannel ECAPA-TDNN. 2022 International Symposium on Advances in Informatics, Electronics and Education (ISAIEE); 2022.

[7]Kusumawati D, Ilham A A, Achmad A. Vgg-16 and Vgg-19 Architecture Models in Lie Detection Using Image Processing. 2022 6th International Conference on Information Technology, Information Systems and Electrical Engineering (ICITISEE); 2022.

[8]Shi ML, Zheng DL, Wu TH, Zhang WJ, Fu RJ, Huang, KL. Small Object Detection Algorithm Incorporating Swin Transformer for Tea Buds. PLOS ONE. 2024;19(3):e0299902.

(10)Comment: Some spelling and grammatical mistakes are found in the article that must be corrected. For example see spelling of “classifiy”, “labeled”, “analysises”, etc., and a grammatical mistake like “…achieving an low Equal Error”, “…combined in pair”, “…are formed. And then, 101 these sub-groups are”

Author Response:

Thank you for highlighting the spelling and grammatical errors in the manuscript. We acknowledge the mistakes you pointed out.

We have thoroughly revised the manuscript to correct these errors in the revised version.

---

## [Decision Letter · Decision Letter 2]

9 Jul 2024

PONE-D-23-39866R2Hierarchical-Concatenate Fusion TDNN for Sound Event ClassificationPLOS ONE

Dear Dr. Liang,

Thank you for submitting your manuscript to PLOS ONE. After careful consideration, we feel that it has merit but does not fully meet PLOS ONE’s publication criteria as it currently stands. Therefore, we invite you to submit a revised version of the manuscript that addresses the points raised during the review process.

We look forward to receiving your revised manuscript.

Kind regards,

K. Martin Sagayam, PhD

Academic Editor

PLOS ONE

Additional Editor Comments:

As per the comment given by the reviewers your paper considered as major revision and the author should address the comments carefully.

Reviewers' comments:

Reviewer's Responses to Questions

**Comments to the Author**

1. If the authors have adequately addressed your comments raised in a previous round of review and you feel that this manuscript is now acceptable for publication, you may indicate that here to bypass the “Comments to the Author” section, enter your conflict of interest statement in the “Confidential to Editor” section, and submit your "Accept" recommendation.

Reviewer #1: All comments have been addressed

Reviewer #4: (No Response)

2. Is the manuscript technically sound, and do the data support the conclusions?

Reviewer #1: Yes

Reviewer #4: (No Response)

3. Has the statistical analysis been performed appropriately and rigorously? 

Reviewer #1: Yes

Reviewer #4: (No Response)

4. Have the authors made all data underlying the findings in their manuscript fully available?

Reviewer #1: Yes

Reviewer #4: (No Response)

5. Is the manuscript presented in an intelligible fashion and written in standard English?

Reviewer #1: Yes

Reviewer #4: (No Response)

6. Review Comments to the Author

Reviewer #1: The author has addressed all the relevant queries. He has clarified the details with proper figures and tables. I recommend that you accept the paper for publication.

Reviewer #4: Review Comments

(1) The abstract is poorly written. Most of the content focuses on the outcome of the proposed model. It should include a little bit of problem background and research objectives as well.

Comment: The objectives in the abstract are still missing.

(2) The literature review should be presented comparatively to show the research gap quantitatively (if possible). For instance, the author can identify some parameters and compare the related models/methodologies based on them in a tabular form.

Comment: The author is unable to present the comparative analysis in a tabular form based on certain parameters. Either it should be added or explained that this type of study cannot be represented in the suggested way.

(3) The research objectives, aims, motivations, and ultimate beneficiaries should be well explained to show the significance of the research contribution.

Comment: It is still missing. The author just explained some of the aspects of the proposed methodology.

(4) It is suggested to add a little bit more description after each figure instead of giving their labels only. For reference, please see figures from numbers 4 to 7.

Comment: Need to be done. The description includes the detail of the figure e.g. its meaning, significance, and relevance to the ongoing discussion.

(5) Future directions should also be discussed to explain the possible future advancements in the underlined research area.

Comment: Future research directions are required instead of deployment details.

(6) The referenced articles are fewer in number to better support the importance and significance of the underlying study. It is recommended to increase the number of referenced articles by adding more related models/methodologies to be compared with the proposed one.

Comment: It is hard to mention a reasonable number of references. But seems that there is still room to increase the scope of the literature review.

7. PLOS authors have the option to publish the peer review history of their article (what does this mean?). If published, this will include your full peer review and any attached files.

Reviewer #1: No

Reviewer #4: No

---

## [Author Response · Author response to Decision Letter 2]

23 Aug 2024

In the revision, we have performed fully addressed the comments made by the reviewers (see the attached "Response to Reviewers" ). We have now revised the manuscript and our main revisions include:

(a)We have added research objectives in abstract section; 

(b)We have supplemented the research objectives, motivations, and aims in the introduction section; 

(c)We have added some relevant references of sound classification model in the introduction section; 

(d)We have added a discussion of more sound classification models in the introduction section; 

(e)We have presented the comparative analysis in a tabular form based on certain parameters in the introduction section;

(f)We have added a new description to the conclusion of our manuscript in future work and beneficiaries.

(g)We have added a little bit more detail description of all figures..

We also provided an itemized response attached to this submission.

---

## [Decision Letter · Decision Letter 3]

17 Oct 2024

Hierarchical-Concatenate Fusion TDNN for Sound Event Classification

PONE-D-23-39866R3

Dear Dr. Liang,

We’re pleased to inform you that your manuscript has been judged scientifically suitable for publication and will be formally accepted for publication once it meets all outstanding technical requirements.

Kind regards,

Anirban Bhowmick, Ph.D.

Academic Editor

PLOS ONE

Additional Editor Comments (optional):

Reviewers' comments:

Reviewer's Responses to Questions

**Comments to the Author**

1. If the authors have adequately addressed your comments raised in a previous round of review and you feel that this manuscript is now acceptable for publication, you may indicate that here to bypass the “Comments to the Author” section, enter your conflict of interest statement in the “Confidential to Editor” section, and submit your "Accept" recommendation.

Reviewer #1: (No Response)

Reviewer #4: All comments have been addressed

2. Is the manuscript technically sound, and do the data support the conclusions?

Reviewer #1: Yes

Reviewer #4: Yes

3. Has the statistical analysis been performed appropriately and rigorously? 

Reviewer #1: Yes

Reviewer #4: Yes

4. Have the authors made all data underlying the findings in their manuscript fully available?

Reviewer #1: Yes

Reviewer #4: Yes

5. Is the manuscript presented in an intelligible fashion and written in standard English?

Reviewer #1: Yes

Reviewer #4: Yes

6. Review Comments to the Author

Reviewer #1: Overall this paper sounds good and it shows that several amendments have been done. However, I suggest following changes that needs attention.

1. The 2nd paragraph of Introduction Section starting at line 15 is too long. It should be split into two paragraphs.

2. You should include few more comparative studies in Table 1 to show more strength of your research.

3. At the end of every figure, figure caption is mixed up with normal text, arrange all figures such that caption remains with the figure and text is on next paragraph.

4. Discuss each layer with bullet points or discuss in a paragraph not each layer on separate paragraph.

5. You have written few sentences only in a paragraph. Check at The fifth layer, The sixth layer, and The seventh layer. Text alignment is big issue in this article. Please make sure your text is written in the paragraph format.

6. Do not leave too much gap/space between paragraphs.

7. It is better to write Network Settings in a tabular form.

8. In Table 5, why sensitivity is 99.6%. What is the reason behind it?

9. Why did you use Kappa evaluation? And what is the significance in this article?

10. In Fig 6. Comparison box plot of Kappa values for different models in sound classification, explain this figure, what does Boxplot shows in this figure.

11. What are you experimental results with different number of epochs or other hyperparameters like learning rate, batch size etc.

12. Future work should include testing with other datasets.

13. Also mention limitations of the current study before conclusion

Reviewer #4: Most of the said changes have been incorporated. This article can be considered for the publication.

7. PLOS authors have the option to publish the peer review history of their article (what does this mean?). If published, this will include your full peer review and any attached files.

Reviewer #1: **Yes: **Muhammad Sajid

Reviewer #4: No

---

## [Editor Report · Acceptance letter]

22 Oct 2024

PONE-D-23-39866R3 

PLOS ONE

Dear Dr. Liang, 

I'm pleased to inform you that your manuscript has been deemed suitable for publication in PLOS ONE. Congratulations! Your manuscript is now being handed over to our production team.

Kind regards, 

on behalf of

Dr. Anirban Bhowmick 

Academic Editor

PLOS ONE